



# Modeling erythemal ultraviolet diffuse fraction

Guadalupe Sanchez[1], Antonio Serrano[1], María Luisa Cancillo[1]

[1]Department of Physics, University of Extremadura, Badajoz, Spain

*Correspondence to*: Guadalupe Sanchez (guadalupesh@unex.es)

**Abstract.-** Although being extremely interesting, the diffuse component of the erythemal ultraviolet irradiance (UVER) is scarcely measured at standard radiometric stations and, therefore, needs to be estimated. This study proposes and compares ten empirical models to estimate the UVER diffuse fraction. These models are inspired on mathematical expressions originally used to estimate total diffuse fraction, but, in this study, they are applied to the UVER case and tested against experimental measurements. In addition to adapting to the UVER range the various independent variables involved in these

models, the total ozone column has been added in order to account for its strong impact on the attenuation of ultraviolet radiation. The proposed models are fitted to experimental measurements and validated against an independent subset. Six models perform notably well, with the best performing model RAU3 showing values of $r^2$ equal to 0.91 and $rRMSE$ equal to 6.1%. The performance achieved by this model is better than those obtained by previous semi-empirical approaches, with the advantage of being entirely empirical and, therefore, needing no additional information from physically-based models.

This study expands previous research to the ultraviolet range, and provides reliable empirical models to accurately estimate the UVER diffuse fraction.

## 1.- Introduction

Data on ultraviolet solar radiation at the Earth's surface is a high priority as it affects many biological, ecological and

photochemical processes [Williamson et al., 2014]. Ultraviolet radiation may have a negative impact on ecosystems such as corals and phytoplankton communities and affect plant growth [Lesser and Farrell, 2004; Zepp et al. 2008; Häder et al., 2011, 2015]. Additionally, ultraviolet radiation is the main factor for degradation of paints and plastics exposed to outdoor conditions [Johnson and McIntyre, 1996; Verbeek et al., 2011].

On the other hand, low doses of ultraviolet radiation are beneficial for human health, particularly for the synthesis of vitamin D3, critical in maintaining blood calcium levels [Webb et al., 1988; Glerup et al., 2000; Holick, 2004]. However, an excessive exposure has adverse consequences such as favoring skin cancer, immune suppression and eye disorders [Diffey, 2004; Heisler, 2010]. The effectiveness of ultraviolet radiation in producing erythema on human skin is usually quantified by the erythemal action spectrum [McKinlay and Diffey, 1987]. The ultraviolet radiation weighted by this action spectrum is

named erythemal ultraviolet radiation (UVER).

Recent studies have shown that, in addition to stratospheric ozone variability, changes in ultraviolet radiation in the last two decades have been influenced by variations in aerosols, clouds, and surface reflectivity [Arola et al., 2003; Herman, 2010]. Significant positive trends in ultraviolet radiation have been detected in different European countries and attributed to a



decrease in cloud cover [Krzyscin et al., 2011; Smedley et al., 2012]. A significant positive trend of 2.1 % per decade in UVER radiation has been detected in the Iberian Peninsula for the period 1985 – 2011 and attributed to aerosol reduction [Roman et al., 2015].

In the framework of the climate change, new variations in ultraviolet irradiance at the Earth's surface are expected for the
next decades as a result of the predicted changes in clouds and aerosols [McKenzie et al., 2007; Bais et al., 2011; Craig et al., 2014]. These variations in ultraviolet irradiance may affect not only the amount but also the diffuse/direct partitioning due to the stronger effectiveness of scattering at shorter wavelengths.

In contrast to the direct component, diffuse ultraviolet irradiance is difficult to block [Utrillas et al., 2010; Kudish et al.,
2011]. For instance, diffuse UVER irradiance under a standard beach umbrella can reach 34% of global UVER irradiance [Utrillas et al., 2007] and up to 60% in a tree shade [Parisi, 2000]. This percentage increases notably with high load of aerosols and presence of clouds, especially in the case of broken-clouds [Alados et al., 2000; Calbó et al., 2005; Esteve et al., 2010]. However, very few studies focus on ultraviolet diffuse irradiance, mainly due to the scarcity in experimental measurements. While global ultraviolet irradiance is commonly registered worldwide, its diffuse component is seldom
measured. Therefore, modeling is a good alternative to partly relieve this scarcity.

There are two main approaches to estimate solar radiation: using physically-based or empirical models. In general, the diffuse component of the radiation field is the magnitude most difficult to estimate, due to the high complexity of the processes involved. This complexity increases for the UV range, where scattering processes are particularly effective. Thus,
physically-based models require a very detailed and accurate description of the composition of the atmosphere, aerosols and clouds to reliably estimate the diffuse radiation. However, this detailed information is often unavailable and, therefore, an empirical approach is needed. Hence, in this paper the empirical approach was preferred because of its simplicity, modest requirements in terms of ancillary data and wide use by the scientific community.

As far as we are aware, only a few studies have applied empirical models to estimate the diffuse solar irradiance in the ultraviolet range [Grant and Gao, 2003; Nuñez et al., 2012; Silva, 2015]. Moreover, the applicability of these studies is limited, since they rely on spectral measurements [Silva, 2015] or require information which is usually unavailable, such as cloud fraction and aerosols properties [Grant and Gao, 2003; Nuñez et al., 2012]. In this context, comprehensive studies focused on the proposal of reliable models based on commonly available data are needed.

In order to contribute to addressing this need, this study aims to propose empirical expressions for modeling hourly UVER diffuse fraction under different sky conditions and to compare their performance against experimental measurements. The proposed expressions will be inspired on the empirical formulae commonly used to estimate the diffuse fraction for total solar irradiance. Several radiometric and geometrical variables will be assessed in order to address their contribution in the
UVER diffuse fraction. Additionally, the total ozone column will be included in the models which are proposed in this



study, due to its essential role for the attenuation of ultraviolet radiation. Finally, the performance of the proposed expressions will be validated against experimental measurements.

## 2.- Instrumentation and data

Data presented here were collected at the radiometric station installed on the roof of the Physics building at the University Campus in Badajoz, Spain. This station is operated by the AIRE research group of the Physics Department of the University of Extremadura. This experimental site is located in south-western Spain (38.9º N; 7.01º W; 199 m a.s.l.). It is characterized by a very dry summer with prevailing cloud-free conditions, leading to UVI values among the highest in Europe, reaching values up to 11. Throughout the rest of the year very different cloud conditions can be found. Clouds are mainly associated

with frontal systems coming from the Atlantic Ocean or to local convective systems. This region is also influenced by different aerosol types such as industrial/urban, mineral and forest fire particles. The mean aerosol optical depth at 440 nm measured at this station is 0.14 and the mean Angström exponent alpha is 1.2 [Obregón et al., 2012]. Extreme aerosol optical depth values higher than 0.3 can be occasionally reached as a result of desert dust intrusions from Sahara Desert (Northern Africa). The large variety of situations that occur during a year guarantees the representativeness of the dataset for

the proposal and assessment of general empirical models. The period of study comprises years 2011 and 2012, which ensures that a large variety of seasonal processes and meteorological conditions are sampled.

The UVER irradiance data used in this study were recorded by two Kipp & Zonen UVS-E-T radiometers with serial numbers #000409 and #080017. The UVS-E-T radiometer measures erythemal ultraviolet irradiance between 280 and 400

nm, following the CIE action spectrum according to ISO 17166:1999 CIE S 007/E-1998 international standard [1998]. This action spectrum was originally proposed by McKinlay and Diffey [1987] to simulate the effectiveness of ultraviolet radiation in producing erythema on human skin. To ensure the reliability of the measurements, the pyranometers of our network are calibrated every two years at "El Arenosillo" Atmospheric Sounding Station of the National Institute for Aerospace Techniques (ESAt/ INTA) in Huelva, Spain (37.10º N, 7.06º W), according to the standard procedure

recommended by the Working Group 4 of the COST Action 726 [Webb et al., 2006; Gröbner et al. 2007; Vilaplana et al., 2009]. The uncertainty of UVER radiometers associated to this calibration procedure is about 5 - 7% [Hülsen and Gröbner, 2007, Vilaplana et al., 2009]. In this study the calibration obtained during the intercomparison campaign held in July 2011 was applied to the period of study (2011-2012).

The data set consists of simultaneous measurements of horizontal global and diffuse UVER irradiance. Thus, while the UVS-E-T radiometer #000409 was installed on a table to measure global UVER irradiance, the UVS-E-T radiometer #080017 was installed on a Kipp & Zonen Solys 2 sun tracker to measure diffuse UVER irradiance. This device prevents the direct solar irradiance to reach the sensor by means of a small ball which continuously projects its shadow on the sensor. Since the portion of the sky obstructed by the shadow ball is negligible, no correction is required for these measurements

[Ineichen et al., 1984].

Global and diffuse UVER measurements were recorded every minute by a Campbell Scientific CR-1000 data logger. One-



minute values for the variables involved in this study such as UVER diffuse fraction, UVER transmissivity, solar zenith angle, and other sun-geometry parameters were calculated and subsequently averaged hourly. This time scale was chosen
since it suitably shows daily variations without been affected by the very fast short-term fluctuations.

Additionally, daily total ozone column (TOC) values as provided by the NASA Ozone Monitoring Instrument (OMI) through their website https://ozoneaq.gsfc.nasa.gov, were used in this study. Since only daily values were available, the ozone amount was assumed to be constant during each day.


**3.- Methodology**

The diffuse component of the solar radiation is usually quantified by the diffuse fraction ($f$) [Liu and Jordan, 1960; Iziomon and Aro, 1998; Utrillas et al., 2007], defined as the ratio between the diffuse ($D$) and the global ($G$) solar irradiances. The diffuse fraction determines the effectiveness of the atmosphere in scattering the incoming radiation. This magnitude is
particularly interesting in the ultraviolet range since scattering is enhanced at shorter wavelengths [Kaskaoutis et al. 2006]. Using ratios of irradiances has the additional advantage of presenting an uncertainty much lower than beam or diffuse irradiances considered separately [Meloni et al., 2006; Badarinath et al., 2007]. Thus, the present study focuses on estimating the UVER diffuse fraction at the Earth's surface, $f_{UVER}$, defined as follows:

$$f_{UVER} = \frac{D_{UVER}}{G_{UVER}}$$

125                                                                                                    (1)

where $D_{UVER}$ and $G_{UVER}$ stand for UVER diffuse and global irradiances respectively.

Although there are very few models for estimating the ultraviolet diffuse fraction [Grant and Gao, 2003; Nuñez et al., 2012; Silva, 2015], several expressions proposed for modeling the diffuse fraction integrated along the complete solar wavelength
interval (termed as total diffuse fraction), can be found in the literature (see, for example, compilations reported by Engerer [2015], and Gueymard and Ruiz-Arias [2016]). These models attempt to describe the absorption and scattering of solar radiation when crossing the atmosphere. Since the mechanisms of absorption and scattering of ultraviolet solar radiation are qualitatively similar to those affecting other solar wavelengths, the models described in this study will be largely based on published models describing the total diffuse fraction. Towards this goal, a complete compilation of models for estimating
total diffuse fraction was performed, the mathematical function and the variables involved were analyzed, and the most suitable models were adapted to the ultraviolet region.

Regarding the independent variables to use, it must be noted that most empirical models for total diffuse fraction are primarily based on the total transmissivity ($k_t$), also named clearness index, as the main factor [Liu and Jordan, 1960; Iqbal,
1980]. Similarly, our proposed models will rely on the UVER transmissivity ($k_{UVER}$), defined as the ratio between the UVER irradiance at the Earth's surface ($G_{UVER}(0)$) and the UVER irradiance at the top of the atmosphere ($G_{UVER}(TOA)$):



$$k_{UVER} = \frac{G_{UVER}(0)}{G_{UVER}(TOA)}$$

(2)

The UVER irradiance at the top of the atmosphere was calculated as follows [Iqbal, 1983]:


$$G_{UVER}(TOA) = S_{UVER} \left(\frac{r_0}{r}\right)^2 \cos(\theta)$$

(3)

where $\theta$ is the solar zenith angle, $(r_0/r)^2$ is the eccentricity correction-factor of the Earth's orbit, and $S_{UVER}$ is the erythemally-weighted solar constant, with an estimated value of 10.031 W/m². The eccentricity correction-factor was calculated using

the Spencer's formula [1971].

Figure 1 shows the relationship between UVER diffuse fraction ($f_{UVER}$) and UVER transmissivity ($k_{UVER}$). A general dependence can be clearly seen though the large scatter suggests the influence of other factors as well. In order to account for this variability, additional magnitudes directly related to the absorption and scattering of radiation in the atmosphere

must be considered.

Other meteorological magnitudes such as temperature, relative humidity and cloud cover could have been included in the models for estimating UVER diffuse fraction. However, these variables are not always available and, therefore, they would limit the applicability of the model. Then, this study focused on models relying on variables commonly available at standard

radiometric stations, such as radiometric magnitudes and sun-geometry parameters directly related to the absorption and scattering of radiation in the atmosphere.

Additionally, in the particular case of the ultraviolet wavelengths, the stratospheric ozone plays a very important role for modulating the radiation that arrives at the earth's surface. Therefore, in principle, the ozone amount must be included in the

models. In order to test its impact on the UVER diffuse fraction ($f_{UVER}$), simulations with SBDART radiative transfer code [Ricchiazzi et al., 1998] were performed sampling different total ozone column values in the range 250 DU to 400 DU. This interval corresponds to the typical range of total ozone column reached in our location along a whole year. In order to analyze the effect due exclusively to changes in ozone, fixed cloud-free conditions and standard atmosphere profiles were considered. The results of the simulations showed that, in addition to showing a large impact on the beam and diffuse

UVER irradiance separately, the total ozone column has a noteworthy impact on their ratio. Thus, for a solar zenith angle of 70º (the largest sampled by our measurements), changing the total ozone column from 250 DU to 400 DU yields a decrease in the UVER diffuse fraction of 4.6%. This clear variation should be considered in order to accurately estimate the UVER diffuse fraction and, therefore, the total ozone column has been included in the empirical models.





### 3.1.- Models


The approaches analyzed in this study correspond to models originally proposed for the total diffuse fraction *f*, but they are here applied for the ultraviolet range. This approach is justified by the fact that the physical processes of absorption and scattering of the UVER radiation are of equal nature to those affecting other solar wavelengths. Therefore, those models that

succeed to describe the total diffuse fraction *f* are, in principle, good candidates for modelling the UVER diffuse fraction $f_{UVER}$.

As mentioned above, total ozone column (TOC) is an essential attenuation factor for the UVER radiation and, therefore, it has been added to the models originally proposed for the total diffuse fraction. This new variable has been included by

adding the a term to each model's mathematical formula. It is to mention that a multiplicative approach consisting in the product of the model's original formula and a power function of TOC, has also been analyzed (not shown). However, the results were essentially the same as those achieved by simply adding a term and, therefore, this latter approach was preferred because of its higher simplicity and parsimony.

### 3.1.1 Reindl et al.: Model REU


The majority of empirical models for estimating the total diffuse fraction represents *f* as a piecewise function of $k_t$ as the main factor [Orgill and Hollands, 1977; Erbs et al, 1982]. This relationship was first proposed by Liou and Jordan (1960) when investigating the relationship between diffuse and global irradiances. Subsequent studies have included other variables

[Bugler, 1977; Iqbal, 1980; Skartveit and Olseth, 1987; Reindl et al., 1990] in an attempt to improve the performance of the original functions.

In contrast, in the ultraviolet range, no piecewise behaviour is detected in the relationship between $f_{UVER}$ and $k_{UVER}$ (Figure 1) and, therefore, a single linear function was proposed for all the range of $k_{UVER}$ values. On the other hand, Figure 1 shows a

large spread in $f_{UVER}$ for fixed values of $k_{UVER}$, suggesting to consider additional factors. Thus, Skartveit and Olseth (1987), and Reindl (1990) and Iqbal (1980) included the solar zenith angle in models for estimating the total diffuse fraction. This factor was added with the goal to account for the enhancement in the Rayleigh scattering as the solar zenith angle increases, mainly in clear days. This effect is even more relevant in the ultraviolet range due to the stronger effectiveness of Rayleigh scattering at shorter wavelengths. Therefore, our first model (named REU) corresponds to that originally proposed by

Reindl et al. (1990) but, in this study, it is applied to the UVER case. Besides, as mentioned before, an additional term containing the total ozone column has been appended. Finally, the model REU proposed is:

$$f_{UVER}^{REU} = a + b \cdot k_{UVER} + c \cdot \cos(\theta) + d \cdot TOC$$

(4)



The name REU of the model stands for: "inspired by **Re**indl et al.' work but, in this study, applied to the **U**VER case". This nomenclature will be applied to the rest of models that are proposed hereinafter in this study.

### 3.1.2 Gonzalez and Calbo: Models GCU

The diffuse fraction shows further variability due to short-term changes in clouds or atmospheric turbidity. Gonzalez and
Calbo [1999] proposed three variables $\Delta_1$, $\Delta_2$ and $\Delta_3$ to account for this variability in the case of total diffuse fraction. In this study, these variables have been applied to the UVER case as follows:

$$\Delta_{UVER,1}=\ln\left(\frac{\sigma}{k_{UVER}}\right)$$

(5)

$$\Delta_{UVER,2}=\ln\left(\frac{1}{(N-1)\overline{k}_{UVER}}\sum_{i=1}^{i=N}\left|k_{UVER(i+1)}-k_{UVER(i)}\right|\right)$$

(6)

$$\Delta_{UVER,3}=\ln\left(\frac{1}{(N-1)\overline{k}_{UVER}}\sum_{i=1}^{i=N}\left|k_{UVER,max}-k_{UVER,min}\right|\right)$$

(7)

where $\sigma$ is the standard deviation, $\overline{k_{UVER}}$ is the mean value of the UVER transmissivity, $k_{UVER,max}$ is the maximum value and
$k_{UVER,min}$ is the minimum value of $k_{UVER}$ for each hour. Although looking similar, these variables mean different approaches to describe the short-term variability in the UVER diffuse fraction. Thus, $\Delta_{UVER,1}$ accounts for intermediate values between the minimum and the maximum, whereas $\Delta_{UVER,3}$ only depends on the extreme values. On the other hand, the fast variations between consecutive measurements are only addressed by variable $\Delta_{UVER,2}$. A logarithmic transformation is applied in Eq. 5, 6 and 7 to avoid values extending over several orders of magnitude and because this transformation increases the effect of
these parameters on the diffuse fraction correlations [Gonzalez and Calbo, 1999]. It should be noted that measurements at a frequency higher than one per hour are needed to calculate these variables.

Similarly to the proposal of Gonzalez and Calbo [1999] for total diffuse fraction, variables $\Delta_{UVER,1}$, $\Delta_{UVER,2}$ and $\Delta_{UVER,3}$ were added to the model REU previously built for UVER diffuse fraction (Eq. 4), resulting in new models that are named GCU1,
GCU2, GCU3:

$$f_{UVER}^{GC1}=a+b\cdot k_{UVER}+c\cdot\cos(\theta)+d\cdot TOC+g\cdot\Delta_{UVER,1}$$

(8)

$$f_{UVER}^{GC2}=a+b\cdot k_{UVER}+c\cdot\cos(\theta)+d\cdot TOC+g\cdot\Delta_{UVER,2}$$

(9)



$$f_{UVER}^{GC3} = a + b \cdot k_{UVER} + c \cdot \cos(\theta) + d \cdot TOC + g \cdot \Delta_{UVER,3}$$
(10)

### 3.1.3 Boland et al. [2001]: Model BOU

Boland et al. [2001] proposed a logistic function to estimate the total diffuse fraction as a function of the total transmissivity.
The logistic functions are S-shaped sigmoid curves where the increase is approximately exponential at the initial stage and, then, the growth slows as saturation begins. This behavior, but with decay, can be useful to describe the dependence of total diffuse fraction ($f$) on $k_t$ [Boland et al., 2001, 2008]. Thus, $f$ decreases as $k_t$ increases, but with a saturation effect towards the clear sky value. This behaviour is also observed in the ultraviolet range and, therefore, the model proposed by Boland et al. has been applied to the UVER case. Additionally, a term including the total ozone column has been added to the exponent of
the exponential function. The resulting model, named BOU, is:

$$f_{UVER}^{BOU} = \frac{1}{1 + \exp(a + b \cdot k_{UVER} + d \cdot TOC)}$$
(11)


### 3.1.4 Ridley et al.: Model RIU

The original expression proposed by Boland et al. [2001] was later expanded by Ridley et al. [2010] to include four additional variables: 1) the solar zenith angle, 2) the apparent solar time *AST*, which accounts for differences in the atmosphere between morning and afternoon, 3) the daily clearness index $K$ calculated as the ratio between the irradiation
accumulated along the whole day at the Earth's surface and its value at the top of the atmosphere, and 4) a variable $\Psi$ to account for the persistence at one hourly scale due to the very slow rate of change in the radiation under cloud-free or overcast skies.

Similarly, the UVER daily clearness index ($K_{UVER}$) and the persistence parameter ($\Psi_{UVER}$) have been calculated for the UVER
case as follows:

$$\Psi_{UVER} = \frac{k_{UVER,i-1} + k_{UVER,i+1}}{2}$$
(12)

$$K_{UVER} = \frac{\sum_{sunrise}^{sunset} G_{UVER}(0)}{\sum_{sunrise}^{sunset} G_{UVER}(TOA)}$$
(13)






where $k_{UVER,i-1}$ and $k_{UVER,i+1}$ are the values before and after each hourly value of $k_{UVER}$. The apparent solar time was calculated according to the Spencer's formulae [Spencer, 1971; Iqbal, 1983].

It has to be noted that, in this study, the variable $\theta$ used by Ridley et al. [2010] has been replaced by $cos(\theta)$ in order to
facilitate the comparison with the other models. Models with $\theta$ and $cos(\theta)$ were tested and showed nearly equal performance (not shown here). Following the Ridley et al.'s expansion, the new model, named RIU, is based on model BOU, where additional terms containing $cos(\theta)$, $AST$, $\Psi_{UVER}$ and $K_{UVER}$ have been included in the exponent of the exponential function:

$$f_{UVER}^{RIU} = \frac{1}{1 + \exp\left(a + b \cdot k_{UVER} + c \cdot \cos(\theta) + d \cdot TOC + g \cdot AST + h \cdot \Psi_{UVER} + j \cdot K_{UVER}\right)} \qquad (14)$$


### 3.1.5 Kuo et al.: Model KUU

Kuo et al. [2014] developed several correlation models aimed to estimate the hourly solar diffuse fraction in Taiwan. They compared four newly proposed models with fourteen models previously available in the literature. As a result of the comparison, they proposed a new model consisting of a multiple linear combination of the same independent variables
included in Ridley et al.'s model. In this study, following Kuo et al.'s suggestion, a model named KUU was built for the UVER case, as follows:

$$f_{UVER}^{KUU} = a + b \cdot k_{UVER} + c \cdot \cos(\theta) + d \cdot TOC + g \cdot AST + h \cdot \Psi_{UVER} + j \cdot K_{UVER} \qquad (15)$$

### 3.1.6 Ruiz-Arias et al.: Models RAU

Similarly to Ridley et al. [2008], Ruiz-Arias et al. [2010] proposed a model for the total diffuse fraction ($f$) based on a sigmoid function of the total transmissivity ($k_t$), but included also the Kasten and Young's relative optical mass [1989] as an additional predictor ($m$). In fact, Ruiz-Arias et al. [2010] proposed three versions of their model corresponding to combinations of $k_t$ and $m$ raised to various powers. Correspondly, in this study, these three models have been applied to the
UVER case and, additionally, a term including the ozone total column has been added to the exponent. Finally, three models named RAU1, RAU2 and RAU3 have been built as follows:

$$f_{UVER}^{RAU1} = A + B \cdot \exp\left(\exp\left(a + b \cdot k_{UVER} + d \cdot TOC\right)\right) \qquad (16)$$


$$f_{UVER}^{RAU2} = A + B \cdot \exp\left(\exp\left(a + b \cdot k_{UVER} + c \cdot m + d \cdot TOC\right)\right) \qquad (17)$$



$$f_{UVER}^{RAU3} = A + B \cdot \exp\left(\exp\left(a + b \cdot k_{UVER} + c \cdot m + d \cdot TOC + g \cdot k_{UVER}^2 + h \cdot m^2\right)\right) \tag{18}$$


### 3.2 Fitting and comparison statistics

This study aims to fit the models to experimental data and subsequently compare their performance using an independent dataset. Towards that aim, the hourly dataset was randomly divided in two subsets: 1) the fitting subset, containing the 75% of data, for fitting the coefficients of the models, and 2) the validation subset, containing the remaining 25% of data, for

model validation and comparison.

The performance of the models proposed to estimate the UVER diffuse fraction was compared using both statistical and graphical tools. The coefficient of determination ($r^2$) and the relative root-mean-square error ($rRMSE$) defined as:

$$r^2 = 1 - \frac{\sum_{i=1}^{i=N}\left(x_i - x_i^*\right)^2}{\sum_{i=1}^{i=N}\left(x_i - \overline{x}\right)^2}$$

315                                                                                                      (19)

$$rRMSE\,(\%) = \frac{100}{\overline{x}}\sqrt{\frac{1}{N}\sum_{i=1}^{i=N}\left(x_i - x_i^*\right)^2}$$

(20)

were used to assess the goodness of fit of the models and their performance. The coefficient of determination is a measure

of the proportion of total variance explained by the model, while the relative root-mean-square error quantifies the difference between modeled and measured values.

Additionally, the Taylor diagram [Taylor, 2001] and the relative differences were used for model comparison. The Taylor diagram provides a concise graphical summary of different aspects of the performance of a model such as the centered root-

mean-square error, the correlation and the standard deviation. On the other hand, the relative residuals between modeled, $x_i^*$ and measured, $x_i$, values are calculated as follows:

$$Residuals\,(\%) = 100 \cdot \frac{x_i - x_i^*}{x_i}$$

(21)

and can be analyzed as a function of the solar zenith angle, the UVER transmissivity, and the UVER diffuse fraction,



## 4. Results and Discussion

Main results of the fitting of each empirical model to the fitting subset are summarized in Table 1. Ordinary least squares
fitting for models REU, GCU1, GCU2, GCU3, BOU, RIU and KUU, and non-linear fitting for models RAU1, RAU2 and
RAU3 have been calculated. The comparison of models GCU1, GCU2 and GCU3 among themselves indicates the better
performance of $\Delta_{UVER,2}$ with respect to $\Delta_{UVER,1}$ and $\Delta_{UVER,3}$ for describing the variability at time periods shorter than one hour.
This primacy of $\Delta_{UVER,2}$ agrees with the case of total diffuse fraction as reported by Gonzalez and Calbó [1999]. Similarly,
models RAU1, RAU2 and RAU3 were compared, finding that RAU3 performs better than the other two. This result is in
line with results reported by Ruiz-Arias et al. [2010] for the total diffuse fraction. Therefore, hereafter, only GCU2 and
RAU3 will be hereinafter considered.

Most of the models performed notably well with $r^2$ higher or equal than 0.83 and $rRMSE$ lower or equal than 8.6%, except
for models BOU and RIU, which perform somewhat worse. This fact confirms the general suitability of different
mathematical functional forms for estimating UVER diffuse fraction and emphasizes the need for comprehensive
comparison studies like the present one.

Subsequently, the various models with their fitted coefficients were applied to the validation subset. The resulting $r^2$ and
$rRMSE$ values are shown also in Table 1. The values are very similar to those obtained for the fitting, indicating no
overfitting effect. The best statistics are achieved by the three-variable model RAU3, with an excellent coefficient of
determination of 0.91 and a low relative root mean squared error of only 6.4%. This model includes $k_{UVER}$, $m$ and $TOC$ as
predictors.

Taylor diagram (Figure 2) confirms the generally good performance achieved by the proposed models, but also identifies
two separate groups: on one hand, models BOU and RIU, and on the other hand, models REU, GCU2, KUU and RAU3, the
latter performing moderately better. It is to note that the worst-performing models BOU and RIU are based on the same
logistic function proposed by Boland et al. [2008]. It can be, therefore, concluded that such functional form is not as
appropriate for the UVER case as those used by the remaining models. Moreover, that worse performance is not improved
even when more variables are included such as in model RIU.

Models REU and KUU completely overlap, indicating that no improvement is achieved when variables $AST$, $\Psi_{UVER}$ and
$K_{UVER}$ are added. Conversely, the variable $\Delta_{UVER,2}$, which was included with the aim to account for the short-term variability,
means a substantial contribution to the better performance achieved by the model CGU2.

In addition to the regression statistics mentioned above, the relative residuals between measured and modeled values were
calculated, and their variation with respect to solar zenith angle, UVER transmissivity and UVER diffuse fraction bands was





analyzed. In order to clearly show the relationship with a particular variable, the relative residuals were averaged by intervals in that variable.

Figure 3 confirms the worse performance achieved by models BOU and RIU. The relative residuals for these two models are the largest among the models proposed in this study. These large residuals occur for low solar zenith angle, high $k_{UVER}$ and low diffuse fraction, which correspond to cloud-free conditions near noontime. In addition, models BOU and RIU's relative residuals show a clear relationship with the three independent variables analyzed, suggesting that their functional form does not properly account for the relationship of $f_{UVER}$ with the solar zenith angle and with $k_{UVER}$.


In contrast, models REU, GCU2, KUU and RAU3 show much smoother patterns, with absolute relative residuals smaller than 5% for almost the entire range of $\theta$ and $k_{UVER}$. Concerning UVER diffuse fraction, these models tend to underestimate for intermediate values and overestimate for high values over 0.8. The model RAU3 is again the preferred model, with absolute relative residuals smaller than 3% except for the lowest values of the UVER diffuse fraction.


Table 2 shows the fitting coefficients for each proposed models. It is important to note that although the functional form can be generally suitable for other locations, the particular values of the coefficients are specific for our local conditions. Therefore, in order to apply the models to other locations, the coefficients should be calculated by fitting to local measurements.


**5.- Conclusions**

This study aims to accurately estimate hourly UVER diffuse fraction at the earth's surface using empirical models. Towards
this goal, ten mathematical expressions are proposed and their performance is compared to experimental measurements. All the empirical models analyzed are based on mathematical expressions originally suggested by Reindl. et al [1990], Gonzalez and Calbo [1999], Boland et al. [2008], Ridley et al. [2010], Kuo [2014], and Ruiz-Arias et al. [2010] for modeling the total diffuse fraction but, in this study, they are applied to the UVER case. Among a complete compilation of formulae used for estimating total diffuse fraction, those models that rely on variables commonly available at standard
radiometric stations are selected. This criterion is applied in order to favor the general applicability of the results of the study. Additionally, a term including the total ozone column is added to account for the important role played by the stratospheric ozone in modulating the ultraviolet radiation that arrives at the earth's surface. As a result, the models REU, GCU1, GCU2, GCU3, BOU, RIU, KUU, RAU1, RAU2 and RAU3 are built, fitted against experimental data and finally validated.


The fitting to experimental measurements revealed a generally good performance of all models except for models BOU and RIU, which perform somewhat worse. It can be said that the proposal of mathematical expressions and variables succeed to



describe the variation in the UVER diffuse fraction. Results indicate that multiple linear combinations and the sigmoid function suggested by Ruiz-Arias [2010] are more suitable for the UVER case than the logistic function proposed by Boland

et al. [2008]. In the case of total integrated radiation, logistic models proved to be useful since they reliably describe the abrupt change shown by the relationship between the total diffuse fraction and the total transmissivity. However, for the UVER measurements that relationship is much smoother and, therefore, the logistic models BOU and RIU provide no improvement with respect to more simple linear models REU, GCU2 and KUU. Conversely, the more complex sigmoid function proposed by Ruiz-Arias et al. [2010] achieves the best fitting statistics.


The fitting results are confirmed by the validation against an independent subset of measurements. The best performing model is RAU3 followed by GCU2, REU and KUU, and finally by RIU and BOU, which perform notably worse, with $r^2$ lower than 0.8 and $rRMSE$ higher than 9%. In particular, the model RAU3 achieves an excellent coefficient of determination of 0.91 and a low relative root mean squared error of only 6.4%. These are very good numbers compared to the only two

approaches for UVER diffuse fraction that, to our knowledge, have been published up to date. Thus, Nunez et al.'s semi-empirical approach applied to Valencia (Spain) achieved an $r^2$ equal to 0.84 [Nuñez et al., 2012], and Silva reported an $r^2$ of 0.79 for his study of Belo Horizonte (Brazil) [Silva, 2015]. It is important to notice that model RAU3 achieved better $r^2$ value while being the only entirely empirical model and, therefore, needing no additional information from physically-based models. This is an important advantage since the latter require detailed information which is often unavailable, limiting their

applicability.

Regarding the residuals, RAU3 is again the best models, with almost all absolute values smaller than 3% and no dependency with $\theta$, $k_{UVER}$, nor $f_{UVER}$. Then, it can be concluded that the proposal of models have succeed in providing empirical models to accurately estimate the UVER diffuse fraction, with the RAU3 model being the preferred one.


This study positively contributes to estimate UVER diffuse irradiance and UVER diffuse fraction in locations where only UVER global irradiance measurements are available. Additionally, the models proposed here can be used to expand time series of UVER diffuse radiation to periods when global but not diffuse UVER irradiance was being measured. In order to assess the general validity of the proposed models, similar research must be conducted in other locations.


**6.- Data availability:** The data analyzed in this study are available from authors upon request (guadalupesh@unex.es).

**7.- Acknowledgements**

This study was partially supported by the research projects CGL2014-56255-C2-1-R granted by the Ministerio de Economía y Competitividad from Spain and by Ayuda a Grupos GR15137 granted by Junta de Extremadura and Fondo Social Europeo (FEDER). Guadalupe Sanchez Hernandez thanks the Ministerio de Economía y Competitividad for the predoctoral FPI





grant BES-2012-054975.

## 8.- References


Alados, I., Foyo-Moreno, I., Olmo, F.J., and Alados-Arboledas, L.: Estimation of photosynthetically active radiation under cloudy conditions, Agric. For. Meteorol., 102, 39–50, 2000.

Badarinath K.V., Kharol S.K., Kaskaoutis D.G., Kambezidis H.D.: Case study of a dust storm over Hyderabad area, India: its impact on solar radiation using satellite data and ground measurements, Sci. Total. Environ., 384, 316-332, 2007.

Bais A.F., Tourpali K., Kazantzidis, A., Akiyoshi, H., Bekki, S., Braesicke, P., Chipperfield, M.P., Dameris, M., Eyring, V., Garny, H., Iachetti, D., Jöckel, P., Kubin, A., Langematz, U., Mancini, E., Michou, M., Morgenstern, O., Nakamura, T.,
Newman, P.A., Pitari G., Plummer, D.A., Rozanov, E., Shepherd, T.G., Shibata, K., Tian, W., and Yamashita, Y.: Projections of UV radiation changes in the 21st century: impact of ozone recovery and cloud effects, Atmos. Chem. Phys., 11, 7533-7545, 2011.

Boland, J., Scott, L., and Luther, M.: Modelling the diffuse fraction of global solar radiation on a horizontal surface, *Environmetrics*, 12, 103–116, 2001.

Boland, J., Ridley, B., and Brown, B.: Models of diffuse solar radiation, Renew. Energy, 33, 575–584, 2008.

Calbó, J., Pages, D., and González, J.A.: Empirical studies of cloud effects on UV radiation: A review, Rev. Geophys., 43, RG2002, doi:10.1029/2004RG000155, 2005.

CIE (1998), Erythema reference action spectrum and standard erythema dose, Vienna, ISO 17166:1999/CIE S007-1998.

De Miguel, A., Bilbao, J., Aguilar, R., Kambezidis, H. D., and Negro, E.: Diffuse solar irradiation model evaluation in the north Mediterranean belt area, Sol. Energy, 70, 143–153, 2001.

Diffey B.L.: Solar ultraviolet radiation effects on biological systems, Phys. Med. Biol., 36, 299-328, 1991.

Diffey, B. L.: Climate change, ozone depletion and the impact on ultraviolet exposure of human skin, Phys. Med. Biol., 49, 1–11, 2004.



Engerer, N.A.: Minute resolution estimates of the diffuse fraction of global irradiance for southeastern Australia, Sol. Energy, 116, 215–237, 2015.


Erbs, D., Klein, S.A., and Duffie, J.A.: Estimation of the diffuse radiation fraction for hourly, daily and monthly average global radiation, Sol. Energy, 28, 293–302, 1982.

Esteve, A. R., Marín, M.J., Tena, F., Utrillas, M.P., and Martínez-Lozano, J.A.: The influence of cloudiness over the values
of UVER in Valencia, Spain, Int. J. Climatol., 30, 127–136, doi:10.1002/joc.1883, 2010.

Glerup, H., Mikkelsen, K., Poulsen, L., Hass, E., Overbeck, S., and Thomsen, J.: Commonly recommended daily intake of vitamin D is not sufficient if sunlight exposure is limited, J. Intern. Med., 247, 260-268, 2000.

Gonzalez, J.A., and Calbo, J.: Influence of the global radiation variability on the hourly diffuse fraction correlations, Sol. Energy, 65, 119–131, 1999.

Grant, R. H., and Gao W.: Diffuse fraction of UV radiation under partly cloudy skies as defined by the Automated Surface Observation System (ASOS), J. Geophys. Res., 108, 4046, doi:10.1029/2002JD002201, 2003.

Gröbner, J., Hülsen, G., Vuilleumier, L., Blumthaler, M., Vilaplana, J.M., Walker, D., and Gil, J.E.: Report of the PMOD/WRC-COST Calibration and Intercomparison of Erythemal radiometers, 2007.

Gueymard C.A., and Ruiz-Arias, J.A.: Extensive worldwide validation and climate sensitivity analysis of direct irradiance
predictions from 1-min global irradiance, Solar Energy, 128, 1–30, 2016.

Häder D.P., Helbling, E.W., Williamson, C.E., and Worrest, R.C.: Effects of UV radiation on aquatic ecosystems and interactions with climate change, Photochem. Photobiol. Sci., 10, 242 -260, 2011.

Häder D.P., Williamson, C.E., Wängberg, S.Ä., Rautio, M., Rose, K.C., Gao, K., Helbling, E.W., Sinha, R.P., Worrest, R.C.: Effects of UV radiation on aquatic ecosystems and interactions with other environmental factors, Photochem. Photobiol. Sci., 14, 108-126, 2015.

Herman, J. R.: Global increase in UV irradiance during the past 30 years (1979–2008) estimated from satellite data, J.
Geophys. Res., 115, doi: 10.1029/2009JD012219, 2010.





Heisler, G. M.: Urban forest influence on exposure to UV radiation and potential consequences for human health, in UV Radiation in Global Climate Change. Measurements, Modeling and Effects on Ecosystems, edited by Gao, W., et al., 331– 369, Tsinghua University Press, Beijing, 2010.


Holick, M. F.: Vitamin D: importance in the prevention of cancers, type 1 diabetes, heart disease and osteoposoris, Am. J. Clin. Nutr., 79, 362–371, 2004.

Hollósy F.: Effects of ultraviolet radiation on plant cells, *Micron*, 33, 179 -197, 2002.


Hülsen, G., Gröbner, J.: Characterization and calibration of ultraviolet broadband radiometers measuring erythemally weighted irradiance, Appl. Opt., 46, 5877–5886, 2007.

Iqbal, M.: Prediction of hourly diffuse solar radiation from measured hourly global radiation on horizontal surface, Sol.
Energy, 24, 491-503, 1980.

Iqbal, M.: An Introduction to Solar Radiation, Academic Press, Ontario, Canada, 1983.

Iziomon M.G. and Aro, T.O.: The diffuse fraction of global solar irradiance at a tropical location, Theor. Appl. Climatol, 61,
525   77-84, 1998.

Johnson B.W., and McIntyre, R.: Analysis of test methods for UV durability predictions of polymer coatings, Prog. Org. Coat., 27, 95 -106, 1996.

Kaskaoutis D.G., Kambezidis, H.D., Jacovides, C.P., Steven, M.D.: Modification of solar radiation components under different atmospheric conditions in the Greater Athens Area, Greece, J. Atmos. Solar Terr. Phys., 68, 1043-1052, 2006.

Kasten, F. and Young, A. T.: Revised optical air mass tables and approximatio formula, *Applied Optics*, 28, 4735–4738, 1989.


Kataria, S., Jajoo, A. and Guruprasad K.N.: Impact of increasing Ultraviolet-B (UV-B) radiation on photosynthetic processes, Journal of Photochemistry and Photobiology, 137, 55-66, 2014.



Krzyścin J.W., Sobolewski, P.S., Jarosławski, J., Podgórski, J., and Rajewska-Więch, B.: Erythemal UV observations at Belsk, Poland, in the period 1976–2008: Data homogenization, climatology, and trends, Acta Geophysica, 59, 155-182, 2011.

Kudish, A. I., Harari M., and Evseev E.G.: The solar ultraviolet B radiation protection provided by shading devices with regard to its diffuse component, Photodermatol. Photoimmunol. Photomed., 27, 236–244, 2011.


Kuo, C.W., Chang, W.C., and Chang, K.C. : Modeling the hourly solar diffuse fraction in Taiwan, Renew. Energy, 66, 56–61, 2014.

Liu B.Y.H., and Jordan R.C.: The inter-relationship and characteristic distribution of direct, diffuse and total solar radiation,
Solar Energy, 4, 1-19, 1960.

McKenzie R.L., Aucamp, P.J., Bais, A.F., Björn, L.O., and Ilyas M.: Changes in biologically-active ultraviolet radiation reaching the Earth's surface, Photochem. Photobiol. Sci., 6, 218 – 231, 2007.

McKinlay A.F. and Diffey B.L. (Passchier W.F. and Bosnajakovic B.F.M.): Human Exposure to Ultraviolet Radiation: Risks and Regulations, Elsevier, Amsterdam, 1987.

Meloni D., di Sarra, A. , Pace, G. , Monteleone, F.: Aerosol optical properties at Lampedusa (Central Mediterranean) - 2. Determination of single scattering albedo at two wavelengths for different aerosol types, Atmos. Chem. Phys., 6, 715-727,
560 2006.

Nuñez, M., Utrillas M.P., and Martinez-Lozano, J.A.: Approaches to partitioning the global UVER irradiance into its direct and diffuse components in Valencia, Spain, J. Geophys. Res., 117, doi:10.1029/2011JD016087, 2012.

Obregón, M.A., Pereira, S., Wagner, F., Serrano, A., Cancillo, M.L., and Silva, A.M.: Regional differences of column aerosol parameters in western Iberian Peninsula, *Atmos. Environ.,* 12, 1 – 10, 2012.

Orgill, J.F., and Hollands K.G.T.: Correlation equation for hourly diffuse radiation on a horizontal surface, Sol. Energy, 19, 357–359, 1977.






Parisi, A. V., Kimlin M.G., Wong, J.C.F., and Wilson, M.: Diffuse component of solar ultraviolet radiation in tree shade , J. Photochem. Photobiol. Biol., 54, 116–120, 2000.

Perez, R., Ineichen, P., Seals, R., and Zelenka, A.: Making full use of the clearness index for parameterizing hourly insolation conditions, Sol. Energy, 45, 111–114, 1990.

Piedehierro A.A., Anton, M. , Cazorla, A., Alados-Arboleda L., and Olmo, F.J.: Evaluation of enhancement events of total solar irradiance during cloudy conditions at Granada (Southeastern Spain), Atmospheric Research, 135-136,1-7, 2014.

Reindl, D.T., Beckman, W.A., and Duffie, J.A.: Diffuse fraction correlations, Sol. Energy, 45, 1–7, 1990.

Ricchiazzi P., Yang, S., Gautier, C., Sowle, D.: SBDART: A research and teaching sortware tool for plane-parallel radiative transfer in the Earth's atmosphere, Bulletin of American Meteorological Society, 79, 2101-2114, 1998.

Ridley, B., Boland, J., and Lauret, P.: Modelling of diffuse solar fraction with multiple predictors, Renew. Energy, 35, 478–483, 2010.

Román R., Bilbao, J. and de Miguel A.: Erythemal ultraviolet irradiation trends in the Iberian Peninsula from 1950 to 2011, *Atmos. Chem. Phys.*, 15, 375–391, 2015.

Ruiz-Arias, J.A., Alsamamra, H., Tovar-Pescador,J., and Pozo-Vazquez, D.: Proposal of a regressive model for the hourly diffuse solar radiation under all sky conditions, Energy Convers. Manag., 51, 881–893, 2010.

Silva, A.A.: The diffuse component of erythemal ultraviolet radiation, Photochem. Photobiol. Sci., 14, 1941-1951, 2015.

Skartveit, A., and Olseth, J.A.: A model for the diffuse fraction of hourly global radiation, Sol. Energy, 38, 271–274, 1987.

Smedley A.R.D., Rimmer, J.S., Moore, D., Toumi, R., and Webb, A.R.: Total ozone and surface UV trends in the United Kingdom: 1979–2008, Int. J Climatol., 32, 338-346, 2012.

Spencer, J. W.: Fourier series representation of the position of the sun, Search, 2, 172, 1971.

Taylor, K.E.: Summarizing multiple aspects of model performance in a single diagram , J. Geophys. Res., 106, 7183–7192,



2001.


Utrillas, M. P., M. J. Marín, Esteve, A.R., Tena, F., Cañada, J., Estellés, V., and Martínez-Lozano J.A.: Diffuse UV erythemal radiation experimental values, *J. Geophys. Res.,* 112, doi:10.1029/2007JD008846, 2007.

Utrillas, M. P., Martınez-Lozano, J.A., and Nuñez, M.: Ultraviolet Radiation Protection by a Beach Umbrella, Photochem.

Photobiol., 86, 449–456, 2010.

Verbeek C.J.R., Hick, T., and Langdon, A.: Degradation as a result of UV radiation of bloodmeal-based thermoplastics, Polym. Degrad. Stab., 96, 515-522, 2011.

Vilaplana, J. M., Serrano, A., Antón, M., Cancillo, M.L., Parias, M., Gröbner, J., Hülsen, G., Zablocky G.,, Díaz, A., and de la Morena, B.A.: COST Action 726 – Report of the "El Arenosillo"/INTA-COST calibration and intercomparison campaign of UVER broadband radiometers, 2007.

Webb, A. R., L. Kline, and Holick, M.F., Influence of season and latitude on the cutaneous synthesis of Vitamin D3:

exposure to winter sunlight in Boston and Edmonton will not promote Vitamin D3 synthesis in human skin, J. Clin. Endocrinol. Metab., 67, 373–378, 1988.

Webb, A., Gröbner J., and Blumthaler, M.: A practical guide to operating broadband instruments measuring erythemally weighted irradiance, Produced by the join efforts of WMO SAG UV and Working Group 4 of the COST-726 Action: Long

Term Changes and Climatology of the UV Radiation over Europe, vol. EUR 22595/WMO, 2006.

Williamson, C.E., Zepp, R. G., Lucas, R.M., Madronich, S., Austin, A.T., Ballaré, C.L., Norval, M., Sulzberger, B., Bais A.F., McKenzie, R.L., Robinson, S.A., Häder, D.P., Paul, N.D., and Bornman, J.F. : Solar ultraviolet radiation in a changing climate, Nat. Clim. Change, 4, 434 – 441, 2014.






**Figures**

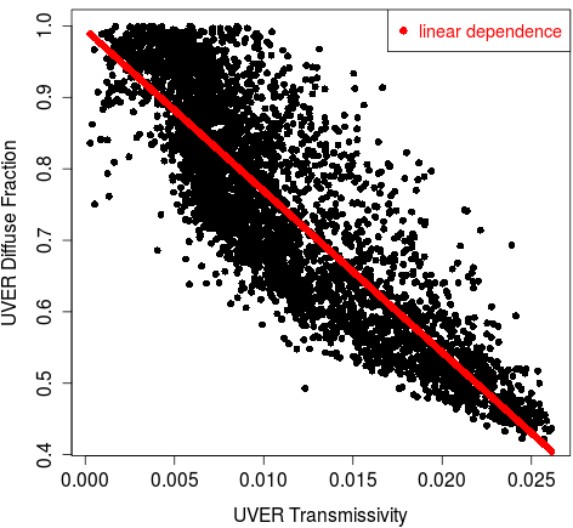

645          Figure 1. UVER diffuse fraction ($f_{UVER}$) versus UVER transmissivity ($k_{UVER}$), and linear fitting.




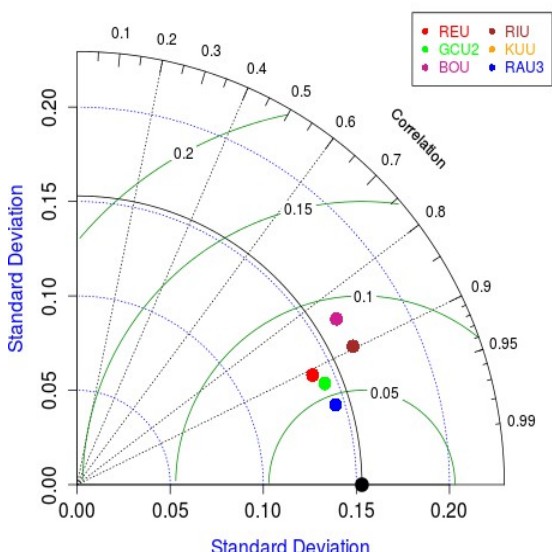

Figure 2. Taylor diagram showing the performance of the models proposed to estimate the diffuse fraction, as compared to experimental measurements.


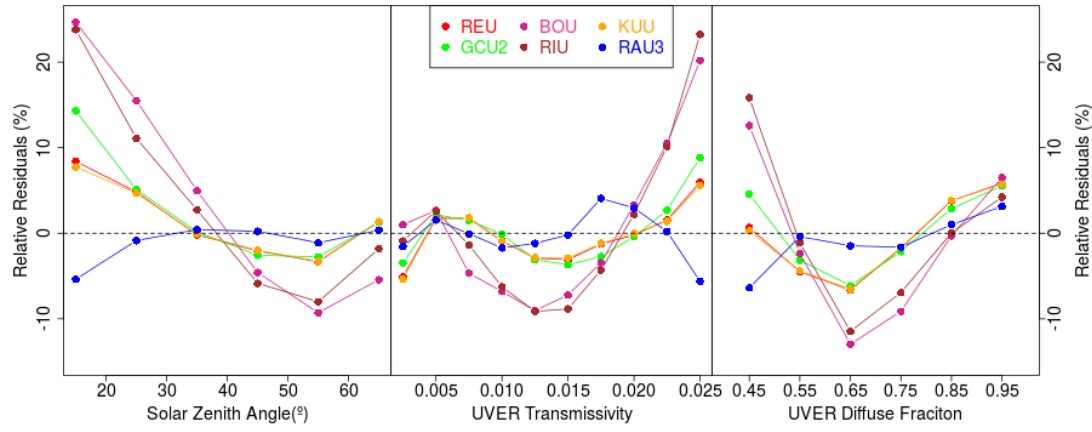

Figure 3. Relative residuals of each UVER diffuse fraction model versus a) solar zenith angle, b) UVER transmissivity, and c) corresponding predicted UVER diffuse fraction values.






Table 1. Coefficient of determination and relative root mean squared error corresponding to the fitting against experimental measurements and the validation of each model.

| Model | Fitting | | Validation | |
|---|---|---|---|---|
| | R² | rRMSE (%) | R² | rRMSE (%) |
| REU | 0.83 | 8.5 | 0.82 | 8.8 |
| GCU2 | 0.86 | 7.7 | 0.85 | 8.0 |
| BOU | 0.66 | 12.0 | 0.68 | 11.8 |
| RIU | 0.78 | 9.8 | 0.78 | 9.7 |
| KUU | 0.83 | 8.6 | 0.82 | 8.8 |
| RAU3 | 0.91 | 6.1 | 0.91 | 6.4 |




Table 2. Empirical fitting coefficients and their corresponding standard error.

| Model | Expression for $f_{UVER}$ |
|---|---|
| REU | $(1.20\pm0.01)+(-35.4\pm0.4)\cdot k_{UVER}+(0.50\pm0.01)\cdot\cos(\theta)+(-1.12\pm0.04)\,x\,10^3\cdot TOC$ |
| GCU2 | $(1.34\pm0.01)+(-28.4\pm0.4)\cdot k_{UVER}+(0.32\pm0.01)\cdot\cos(\theta)+(-1.12\pm0.03)\,x10^{-3}\cdot TOC+(0.032\pm0.001)\cdot\Delta_{UVER,2}$ |
| BOU | $\dfrac{1}{1+\exp\left((-3.7\pm0.1)+(146\pm2)\cdot k_{UVER}+(2.4\pm0.4)\,x\,10^{-3}\cdot TOC\right)}$ |
| RIU | $\dfrac{1}{1+\exp\left((-5.0\pm0.1)+(71\pm14)\cdot k_{UVER}+(-4.2\pm0.1)\cdot\cos(\theta)+(9.9\pm0.4)\,x10^{-3}\cdot TOC+(-5.0\pm10)\,x10^{-3}\cdot AST+(-46\pm14)\cdot\Psi_{UVER}+(187\pm26)\cdot K_{UVER}\right)}$ |
| KUU | $(1.23\pm0.01)+(-59\pm13)\cdot k_{UVER}+(0.47\pm0.01)\cdot\cos(\theta)+(-1.17\pm0.04)\,x10^{-3}\cdot TOC+(2.3\pm1.4)\,x10^{-3}\cdot AST+(23\pm13)\cdot\Psi_{UVER}+(-11\pm2)\cdot K_{UVER}$ |
| RAU3 | $(0.50\pm0.01)+(0.51\pm0.01)\cdot\exp\left(\exp\left((-23.4\pm0.6)+(788\pm20)\cdot k_{UVER}+(9.1\pm0.3)\cdot m+(-13.3\pm0.4)\,x10^3\cdot k_{UVER}^2+(-1.61\pm0.05)\cdot m^2+(1.76\pm0.05)\,x10^{(-2)}\cdot TOC\right)\right)$ |