# Peer review of "Modeling the erythemal surface diffuse irradiance fraction for Badajoz, Spain"

_Atmospheric Chemistry and Physics, 2017_

## Referee Comment (RC1) · Anonymous Referee #1 · 7 Aug 2017

The manuscript describes adoption of models of the total solar diffuse irradiance to the erythemal weighted ultraviolet diffuse irradiance. Several models are proposed and tested against measured data for the location of Badajoz, Spain.

The manuscript is well-organised and include description and verification of new/adopted model results. The English language may be improved in places. It is recommended to have the manuscript proof-read by a fluent English speaker.

Specific suggestions for improvements are given below.

- **Page 1, line 7**: The title is rather general while the topic of the manuscript is rather limited and does not at all cover what is promised in the title. The words "surface" and "irradiance" should be included in the title and also preferable the

[Figure]

location for which the study was made. A suggestion for the title is "Modeling the erythemal surface diffuse irradiance fraction for Badajoz, Spain".

- **Page 1, line 7**: "inspired on" → "inspired from".

- **Page 1, line 12**: The RAU3 acronym has no meaning unless the full manuscript is read. Maybe rather write "the best performing model (RAU3) is based on a model proposed by Ruiz-Arias et al. [2010] and shows values . . . ".

- **Page 1, lines 13-14**: Maybe write this sentence as: "The performance achieved by this entirely empirical model is better than those obtained by previous semi-empirical approaches, and therefore needs no additional information from other physically-based models."

- **Page 2, line 40**: Craig et al. reference missing in References.

- **Page 2, line 41**: It is the variations in the clouds and aerosols that affect the diffuse/direct partitioning and not the variations in the ultraviolet irradiance. Please clarify this sentence.

- **Page 2, line 55**: Please include a reference and/or an example of a physically-based model.

- **Page 2, line 58**: You state that empirical model have a "wide use by the scientific community". May you please provide some references reflecting the wide use? In the next sentence, line 60, it is stated that "few studies have applied empirical models . . . " contradicting your claim on line 58.

- **Page 2, line 69**: "contribution in" → "contribution to".

- **Page 3, line 78**: The acronym UVI is introduced without explaining what it is and how it is related to UVER. It is not used later in the manuscript. Please explain UVI.

- **Page 3, lines 84-86**: It may be argued that at least two important processes that largely affect UVER are not included in the dataset used here. One is the effect of snow on the surface which significantly changes UVER. The other is altitude. Both these processes affect both total UVER and the direct/diffuse ratio. It should at least be mentioned that these processes are not included in the dataset and that the proposed models thus have not been tested for these processes.

- **Page 4, lines 109-110**: On line 84 you argue that the dataset includes a "large variety of situations", then you reduce the variety significantly by hourly averaging this dataset. Please provide a sound justification that the reduced dataset also includes a "large variety of situations". And it should be shown that the reduced dataset with hourly resolution produces similar results as the original higher time resolution dataset.

  Should not the proposed model also account for short-term fluctuations? If not, why are not these important for users of these type of models? If short-term fluctuations are not included, this should be stated as a limitation of the model.

  Did you hourly average the solar zenith angle? It would seem more appropriate to average the cosine of the solar zenith angle. May you please comment on this?

- **Page 4, line 119**: "fraction determines the" → "fraction describes the".

- **Page 5, line 149**: It should be mentioned that the solar constant is not a constant and that it may vary over the solar cycle, especially in the UV, see for example Lean et al. [1992] and Kopp and Lean [2011]. Please mention the uncertainties in the estimated erythemally-weighted solar constant due to the variations in the solar constant.

- **Page 5, line 160**: "radiometric magnitudes" → "radiometric quantities".

- **Page 6, line 185**: What is meant by "adding the a term"? Please clarify sentence.

- **Page 6, line 203**: "in clear" → "on clear".

- **Page 10, line 319**: The description of $x_i$ and $x_i^\star$ should follow immediately after Eqs. (19) and (20) and not later in a different paragraph, lines 325-326.

- **Page 11, line 334**: When reading this sentence I thought results from all models would be included in Table 1. However, the results for GCU1, RAU1 and RAU2 are excluded. It is stated that results for these models were calculated. Hence please include these results in Table 1 to make the manuscript complete.

- **Page 12, Conclusions**: It should be mentioned in the Conclusions that the models have not been tested for high albedo (snow) conditions nor for high altitudes. Also, the testing have been limited to solar zenith angles less than $70°$. Hence the models may not be suitable for the large solar zenith angles encountered at high latitudes.

- **Page 22, lines 485-486**: It should be stated in the caption that the black dot is the observation. Also describe what value the green lines represents.

- **Page 23, Table 2, caption**: This tables does not only list the "Empirical fitting coefficients and their corresponding standard error" as stated in the caption, but the full functional form of the models. This should be mentioned in the caption. It should also be mentioned in the caption that these coefficients are only applicable to the Badajoz, Spain, site. This to avoid that others misuse these equations for other locations.

References

- Kopp, Greg and Lean, Judith L., A new, lower value of total solar irradiance: Evidence and climate significance, Geophysical Research Letters, 38, 1, L01706, DOI:10.1029/2010GL045777, 2011.

- Lean, Judit, Michael VanHoosier, Guenter Brueckner, Dianne Prinz, Linton Floyd and Kenneth Edlow, SUSIM/UARS observations of the 120 to 300 nm flux variations during the maximum of the solar cycle: inference for the 11-year cycle, Geophysical Research Letters, 19, 2203-2206, 1992.
* * *

---

## Referee Comment (RC2) · Anonymous Referee #2 · 9 Aug 2017

General comments:

This paper presents estimates of UVER diffuse fraction (ratio between UVER diffuse radiation and global UVER radiation) using several empirical models based on models for total radiation found in the literature. The authors use measurements of UV radiometers to determine the fitting coefficients of the models. Further, they check the derived expressions with other radiometer measurements. The aim of their work is to provide an efficient model of UVER diffuse fraction enabling deriving the UVER diffuse radiation from UVER global radiation measurements.

This kind of study is useful since UVER diffuse radiation measurements are not so

frequent as global UVER radiation measurements.

Specific comments:

- p. 1, line 5: The authors say "Although being extremely interesting...", they must explain why the diffuse component of UVER is so interesting.

- p. 1, lines 19-28: The sentence in lines 25-26 concerns beneficial effect of UV, it is written between negative effects in lines 20-23 and 26-29. I suggest all beneficial effects be gathered, the same for adverse effects.

- p. 4, line 112: The authors don't give any detail about the processing of Kipp & Zonen measurements. For ex. which TOC is used ? Is it the same as that included in the models which comes from OMI?

- p. 4, line 139: Give the definition of the total transmissivity, kt. Currently, the definition can only be guessed after reading the UVER transmissivity definition in lines 140-141.

- p. 4, line 141: I believe that GUVER(0) is the same as GUVER in Equation (1) and line 126. Please, use the same writing.

- p. 5, Equation (3): r0 and r should be defined separately, not only via their ratio in line 148.

- p. 5, lines 152-155: Figure 1 shows the relationship between UVER diffuse fraction (fUVER) and UVER transmissivity (kUVER). If I understand correctly the first one is derived from measurements of the K&Z radiometers (GUVER(0) and DUVER(0)) and the second one is derived from measurements of GUVER(0) and from a computation (Eq. 3). If it is true, the authors should mention all that (see also another comment below).

- p. 5, line 156: What does "meteorological magnitudes" mean ? Is it not rather "meteorological quantities" or "meteorological parameters" ?

- p. 5, line 156-157: I don't understand why the authors state that other parameters

ACPD
"could have been included in the models for estimating UVER diffuse fraction". As I told just above concerning Fig. 1, it sounds to me that fUVER was derived from measurements, not from modeling. I am rather confused, so the authors should clarify all that.

- p. 5, line 160: Replace "radiometric magnitudes" with "radiometric quantities" or "radiometric parameters".

- p. 7, Eq. (6) and (7): Define N.

- p. 7, line 224: The authors say "where sigma is the standard deviation", they should precise of what it is the standard deviation (of kUVER).

- p. 10, Eq. (19) and (20): xi and xi\* must be defined just after the equations, not much later in lines 325-326.

- p. 10, line 320: What does "total variance " mean ?

- p. 11, line 334: Explain what "ordinary" means.

- p. 11, lines 334-336: The authors must explain why they have chosen different fittings (ordinary least squares and non-linear) for two groups of models.

- p. 12, lines 367-368 and Fig.3: The relative residuals are averaged by intervals of each variable, these intervals should be specified. In Fig. 3 the caption should mention "Mean relative residuals of each UVER diffuse fraction...". Moreover dispersion bars around each mean are needed.

- p. 12, line 381-382: The authors state "... although the functional form can be generally suitable for other locations...", did they check that ? If yes they must give examples, if not they must reconsider their statement.

Technical corrections:

- p. 4, line 110: Replace "without been affected" with "without being affected".
- p. 6, line 184-185: Replace "This new variable has been included by adding the a term..." with "This new variable has been included by adding a term" ("the" has been removed).

- p. 7, Eq. (8): In the left hand side term replace the superscript GC1 with GCU1.
- p. 7, Eq. (9): In the left hand side term replace the superscript GC2 with GCU2.
- p. 8, Eq. (10): In the left hand side term replace the superscript GC3 with GCU3.
- p. 9, line 291: "Ridley et al. [2008]"  $\rightarrow$  "Ridley et al. [2010]".
- p. 13, 422: "models"  $\rightarrow$  "model".
- The reference "Arola et al., 2003" cited p. 1, line 33 is missing.
- The reference "Craig et al., 2014" cited p. 2, line 40 is missing.

- Fig. 2: It is difficult to distinguish the various colors red, magenta and brown. Moreover I cannot see the yellow dot. I suggest making different symbols of various colors.

- Fig 3: Replace the x-axis caption of the right plot "Fraciton" with "Fraction". Replace also "Relative residuals" with "Mean Relative residuals" on the y-axes.

- Table 1: The authors should add the number of cases for fitting and validation for each model.

---

## Author Comment (AC1) · 27 Aug 2017

**Response to Referee #1**

We thank the referee for all the valuable comments that have improved this manuscript. As a result, more information and analyses have been included. Additionally, following the referee's suggestion, the writing has been improved and the grammar mistakes have been corrected. Please see below our point-by-point replies to the specific comments, with the referee's comments in black colour and our corresponding replies colored in orange.

**Comment 1 (Page 1, line 7):** The title is rather general while the topic of the manuscript is rather limited and does not at all cover what is promised in the title. The words "surface" and "irradiance" should be included in the title and also preferable the location for which the study was made. A suggestion for the title is "Modeling the erythemal surface diffuse irradiance fraction for Badajoz, Spain".

**Response 1:** Following the referee's suggestion the title has been rewritten as: "Modeling the erythemal surface diffuse irradiance fraction for Badajoz, Spain"

C2 (Page 1, line 7): "inspired on"  $\rightarrow$  "inspired from".

**R2:** The text has been corrected (line 9)

**C3 (Page 1, line 12):** The RAU3 acronym has no meaning unless the full manuscript is read. Maybe rather write "the best performing model (RAU3) is based on a model proposed by Ruiz-Arias et al. [2010] and shows values . . . ".

**R3:** The text has been rewritten following this suggestion (line 13-14).

**C4 (Page 1, lines 13-14):** Maybe write this sentence as: "The performance achieved by this entirely empirical model is better than those obtained by previous semi-empirical approaches, and therefore needs no additional information from other physically-based models."

R4: The text has been rewritten following the referee's suggestion (lines 14-16).

C5 (Page 2, line 40): Craig et al. reference missing in References.

**R5:** The correct reference is: Williamson et al. [2014] (line 40).

This reference was included in the reference list of the original manuscript as:

Williamson, C., R. Zepp, R. Lucas, S. Madronich, A. Austin, C. Ballare, M. Norval, B. Sulzberger, A. Bais, R. McKenzie, S. Robinson, D. Häder, N. Paul, and J. Bornman. Solar ultraviolet radiation in a changing climate, *Nature Climate Change*, 4, 434-441, 2014.

C6 (Page 2, line 41): It is the variations in the clouds and aerosols that affect the diffuse/direct partitioning and not the variations in the ultraviolet irradiance. Please clarify this sentence.

**R6:** This sentence has been rewritten as follows (line 40):

"These variations in clouds and aerosols may affect not only the amount but also the diffuse/direct partitioning due to the stronger effectiveness of scattering at shorter wavelengths."

C7 (Page 2, line 55): Please include a reference and/or an example of a physically-based model.

**R7:** Models libRadtran [Mayer and Kylling, 2005], SBDART [Ricchiazzi et al., 1998] and TUV [Madronich and Flocke, 1997] have been mentioned in line 53-54. Their corresponding references have been also added to the reference list.

**C8** (Page 2, line 58): You state that empirical model have a "wide use by the scientific community". May you please provide some references reflecting the wide use? In the next sentence, line 60, it is stated that "few studies have applied empirical models . . . " contradicting your claim on line 58.

**R8:** Lines 53-70 have been rewritten in order to have a clearer text and some references have been included. The final text is as follows (lines 56-62) :

"Hence, in this paper the empirical approach was preferred because of its simplicity and modest requirements in terms of ancillary data. The empirical approach has been widely use by the scientific community to estimate the diffuse component in the total solar spectrum [Orgill and Hollands, 1977; Iqbal, 1983; Reindl et al., 1990; Gonzalez and Calbo, 1999; De Miguel et al., 2001; Boland et al., 2008; Ridley et al., 2010; Ruiz-Arias, 2010; Engerer, 2015].

In the particular case of the UV range, the complexity in modeling the diffuse component increases due to the higher effectiveness of the Rayleigh scattering. As far as we are aware, ..."

C9 (Page 2, line 69): "contribution in"  $\rightarrow$  "contribution to".

**R9:** The text has been corrected.

**C10 (Page 3, line 78):** The acronym UVI is introduced without explaining what it is and how it is related to UVER. It is not used later in the manuscript. Please explain UVI.

**R10:** Since UVER is explained in the next paragraph and we are here describing the climate general conditions, it has been prefered not to mention the UVI, and rewrite the text as follows (line 82):

"... leading to noon irradiance values among the highest in Europe."

**C11 (Page 3, lines 84-86):** It may be argued that at least two important processes that largely affect UVER are not included in the dataset used here. One is the effect of snow on the surface which significantly changes UVER. The other is altitude. Both these processes affect both total UVER and the direct/diffuse ratio. It should at least be mentioned that these processes are not included in the dataset and that the proposed models thus have not been tested for these processes.

**R11:** Following the referee's comment, the text has been rewritten to arrive at (lines 89-95):

"The period analyzed in this study comprises years 2011 and 2012, which ensures that a large variety of seasonal processes and meteorological conditions are sampled. The large variety of sun-geometry and meteorological situations that occur during a year guarantees the representativeness of the dataset for the proposal and assessment of empirical models for our location. However, it must be mentioned that snow and altitude are additional factors that have not been considered in this study. They are not represented in the dataset and the proposed models have not been tested for the processes they involve. These factors can significantly affect total UVER and the direct/diffuse ratio and, therefore, should be included for high and snowed locations."

**C12 (Page 4, lines 109-110):** On line 84 you argue that the dataset includes a "large variety of situations", then you reduce the variety significantly by hourly averaging this dataset. Please provide a sound justification that the reduced dataset also includes a "large variety of situations". And it should be shown that the reduced dataset with hourly resolution produces similar results as the original higher time resolution dataset. Should not the proposed model also account for short-term fluctuations? If not, why are not these important for users of these type of models? If short-term fluctuations are not included, this should be stated as a limitation of the model. Did you hourly average the solar zenith angle? It would seem more appropriate to average the cosine of the solar zenith angle. May you please comment on this?

**R12:** Line 84 refers to the large variety of sun-geometry and meteorological situations that are sampled by the two year period of study. By including two years of hourly data, the diurnal and annual cycles are suitably described. The text has been rewritten as follows:

"The period of study comprises years 2011 and 2012, which ensures that a large variety of seasonal processes and meteorological conditions are sampled. The large variety of sun-geometry and meteorological situations that occur during a year guarantees the representativeness of the dataset for the proposal and assessment of empirical models for our location."

Additionally, the selection of the hourly scale has been justified as follows (lines 123-129):

" In this study, hourly data has been used similarly similarly to the majority of previous studies [Reindl et al., 1990; González and Calbó, 1999; Boland et al., 2001 and 2008; Ridley et al., 2010, Ruiz-Arias et al., 2010]. According to Ruiz-Arias [2010], while random errors are much lower than shorter intervals, it offers an appropriate agreement between data availability and the inherent solar radiation temporal variability. Thus, this temporal frequency is the one used by many applications, such as house energy ratings scheme software [Boland et al., 2001]. As a consequence, most of the statistical models are based on the hourly interval of the solar radiation data [Ruiz-Arias 2010, Gueymard and Ruiz-Arias, 2016]."

Additionally, in order, three parameters proposed by Gonzalez and Calbo [1999] to account for the short-term fluctuations and another two proposed by Ruiz-Arias et al. [2010] have been included in some models. This way, the relevance of the short-term scales can be tested by comparing models with and without these parameters. The results of this comparison have been included as follows (lines 359-371):

"Main results of the fitting of each empirical model to the fitting subset are summarized in Table 1. Ordinary least squares fitting for models REU, GCU1, GCU2, GCU3, BOU, RIU and KUU, and non-linear fitting for models RAU1, RAU2 and RAU3 have been calculated. As mentioned in Section 3, some models involve parameters accounting for the short-term fluctuation. In particular, models based on Gonzalez and Calbo [1999], that is, GCU1, GCU2 and GCU3, include parameters  $\Delta_{UVER,1}$ ,  $\Delta_{UVER,2}$ ,  $\Delta_{UVER,3}$ , respectively. In the case of model RAU3, inspired from Ruiz-Arias et al. [2010], the parameters  $kt^2$  and  $m^2$  are introduced to account for the short-term variability. Results in Table 1 show that model GCU2, with a short-term variability parameter, presents a better performance than model REU, which has a same functional form without a short-term variability parameter. Conversely, models GCU1 and GCU3 do not show a better performance than REU despite to include short-term variability parameters. This primacy of  $\Delta_{UVER,2}$  agrees with the case of total diffuse fraction as reported by Gonzalez and Calbó [1999]. In the case of models inspired from Ruiz-Arias et al. [2010], RAU1, RAU2 and RAU3, a notable improvement is observed in the values of  $R^2$  and rRMSE when the parameters  $kt^2$  and  $m^2$  are introduced in RAU3. This result is in line with results reported by Ruiz-Arias et al. [2010] for the total diffuse fraction. Therefore, hereafter, only GCU2 and RAU3 will be hereinafter considered." Regarding the averaging, it was indeed cos(SZA) what was averaged and not SZA. The text has been rewritten to clarify it as follows (lines 120-123):

"Global and diffuse UVER measurements were recorded every minute by a Campbell Scientific CR-1000 data logger. Based on these data and the time of each measurement, a one-minute dataset consisting in the UVER diffuse fraction, UVER transmissivity, relative optical mass and cosine of the solar zenith angle was built. Subsequently these quantities were averaged hourly."

C13 (Page 4, line 119): "fraction determines the"  $\rightarrow$  "fraction describes the".

R13: The text has been corrected.

**C14 (Page 5, line 149):** It should be mentioned that the solar constant is not a constant and that it may vary over the solar cycle, especially in the UV, see for example Lean et al. [1992] and Kopp and Lean [2011]. Please mention the uncertainties in the estimated erythemally-weighted solar constant due to the variations in the solar constant.

**R14:** This point has been included in the text as follows (lines 168-172):

"SUVER is the erythemally-weighted solar constant, with an estimated value of 10.031 W/m2. It must be mentioned that the solar constant may vary over the solar cycle, mainly in the very short UV wavelengths [Lean et al, 1992; Kopp and Lean, 2011]. In the case of the erythemally-weighted irradiance at surface, the wavelength interval of interest starts at 290 nm and, therefore, the variation is lower than 1% [Floyd et al, 2002; DeLand and Cebula 2012; Yeo et al, 2015]."

C15 (Page 5, line 160): "radiometric magnitudes"  $\rightarrow$  "radiometric quantities".

**R15:** The text has been corrected.

C16 (Page 6, line 185): What is meant by "adding the a term"? Please clarify sentence.

R16: The text has been corrected. It should have said: "... adding a term...".

C17 (Page 6, line 203): "in clear"  $\rightarrow$  "on clear".

**R17:** The text has been corrected.

**C18 (Page 10, line 319):** The description of xi\* and xi should follow immediately after Eqs. (19) and (20) and not later in a different paragraph, lines 325-326.

**R18:** The definition of xi\* and xi has been now written just after Eqs. (19) and (20).

**C19 (Page 11, line 334):** When reading this sentence I thought results from all models would be included in Table 1. However, the results for GCU1, RAU1 and RAU2 are excluded. It is stated that results for these models were calculated. Hence please include these results in Table 1 to make the manuscript complete.

**R19:** The results of the statistics R2 and rRMSE for models GCU1, GCU3, RAU1 and RAU2 have been included in Table 1.

**C20 (Page 12, Conclusions):** It should be mentioned in the Conclusions that the models have not been tested for high albedo (snow) conditions nor for high altitudes. Also, the testing have been limited to solar zenith angles less than 70°. Hence the models may not be suitable for the large solar zenith angles encountered at high latitudes.

R20: This points has been mentioned in the Conclusions section as follows (157-164):

"It should be mentioned that factors affecting the UVER diffuse fraction such as the altitude or surface albedo have not been tested in this study. Moreover, since only solar zenith angles below 70° have been considered, the models may not be suitable for the large solar zenith angles encountered at high latitudes. Therefore, similar research must be conducted to assess the general validity of the proposed models and/or establish their possible adaptation to other locations. Although these results apply mainly to regions with similar characteristics than those analyzed in this study, the methodology and comparisons described in this paper can be used to develop similar analysis for other locations."

**C21 (Page 22, lines 485-486):** It should be stated in the caption that the black dot is the observation. Also describe what value the green lines represents.

**R21:** This information has been included in the caption of Figure 2 as follows:

"Figure 2. Taylor diagram showing the performance of the models proposed to estimate the diffuse fraction, as compared to experimental measurements. This diagram summarizes different aspects of the performance of a model such as the centered root-mean-square error (green lines), the correlation and the standard deviation with respect to the reference data set (black dot)."

**C22 (Page 23, Table 2, caption):** This tables does not only list the "Empirical fitting coefficients and their corresponding standard error" as stated in the caption, but the full functional form of the models. This should be mentioned in the caption. It should also be mentioned in the caption that these coefficients are only applicable to the Badajoz, Spain, site. This to avoid that others misuse these equations for other locations.

**R22:** This information has been included in the caption of Table 2 as follows:

"Table 2. Functional form of the models and empirical fitting coefficients with their corresponding standard error for Badajoz, Spain."

---

## Author Comment (AC2) · 27 Aug 2017

**Response to Referee #2**

We thank the referee for all the valuable comments that have improved this manuscript. As a result, more information and analyses have been included. Additionally, following the referee's suggestion, the writing has been improved and the grammar mistakes have been corrected. Please see below our point-by-point replies to the specific comments, with the referee's comments in black colour and our corresponding replies colored in blue.

**Comment 1 ( p. 1, line 5):** The authors say "Although being extremely interesting. . .", they must explain why the diffuse component of UVER is so interesting.

**Response 1:** This sentence has been replaced by (line 6):
*"Despite its important role on the human health and numerous biological processes, the diffuse component ..."*

**C2 (p. 1, lines 19-28):** The sentence in lines 25-26 concerns beneficial effect of UV, it is written between negative effects in lines 20-23 and 26-29. I suggest all beneficial effects be gathered, the same for adverse effects.

**R2:** Following the referee's comment, the text has been re-ordered as follows (lines 20-28):

*"Low doses of ultraviolet radiation are beneficial for human health, particularly for the synthesis of vitamin D3, critical in maintaining blood calcium levels [Webb et al., 1988; Glerup et al., 2000; Holick, 2004]. However, an excessive exposure has adverse consequences such as favoring skin cancer, immune suppression and eye disorders [Diffey, 2004; Heisler, 2010]. The effectiveness of ultraviolet radiation in producing erythema on human skin is usually quantified by the erythemal action spectrum [McKinlay and Diffey, 1987]. The ultraviolet radiation weighted by this action spectrum is named erythemal ultraviolet radiation (UVER). Additionally, ultraviolet radiation may have a negative impact on ecosystems such as corals and phytoplankton communities and affect plant growth [Lesser and Farrell, 2004; Zepp et al. 2008; Häder et al., 2011, 2015]. It is also the main factor for degradation of paints and plastics exposed to outdoor conditions [Johnson and McIntyre, 1996; Verbeek et al., 2011]."*

**C3 (p. 4, line 112):** The authors don't give any detail about the processing of Kipp & Zonen measurements. For ex. which TOC is used ? Is it the same as that included in the models which comes from OMI?

**R3:** Additional information about the calibration process of the UVER radiometers has been included as follows (lines 101-111):

*"To ensure the reliability of the measurements, the pyranometers of our network are calibrated every two years at "El Arenosillo" Atmospheric Sounding Station of the National Institute for Aerospace Techniques (ESAt/ INTA) in Huelva, Spain (37.10° N, 7.06° W), according to the standard procedure recommended by the Working Group 4 of the COST Action 726 [Webb et al., 2006; Gröbner et al. 2007; Vilaplana et al., 2007]. This calibration procedure involves both laboratory characterization and outdoors intercomparison against a reference instrument, in our case, the QASUME unit belonging to the PMOD/WRC. The relative angular and spectral response functions are measured in the laboratory and integrated in a calibration matrix which depends on the solar zenith angle and the total ozone column (TOC). The uncertainty of UVER radiometers associated to this calibration procedure is about 5 - 7% [Hülsen and Gröbner, 2007, Vilaplana et al., 2007]. In this study the calibration obtained during the intercomparison campaign held in July 2011 was applied to the period of study (2011-2012). The calibration matrix has been applied using the TOC values provided by the NASA Ozone Monitoring Instrument (OMI) for our location."*

**C4 (p. 4, line 139):** Give the definition of the total transmissivity, kt. Currently, the definition can only be guessed after reading the UVER transmissivity definition in lines 140-141.

**R4:** Following the referee's comments, the text has been rewritten as follows (lines 156-160):

*"Regarding the independent variables to use, it must be noted that most empirical models for total diffuse fraction are primarily based on the total transmissivity ($k_t$), also named clearness index, as the main factor [Liu and Jordan, 1960; Iqbal,1980]. This quantity is defined as the ratio between the total irradiance at the Earth's surface (G (0)) and the total irradiance at the top of the atmosphere (G(TOA)). In this study, our proposed models will rely on the UVER transmissivity ($k_{UVER}$), defined analogously to the total transmissivity but applied to the UVER irradiance as follows:"*

**C5 (p. 4, line 141):** I believe that GUVER(0) is the same as GUVER in Equation (1) and line 126. Please, use the same writing.

**R5:** As the referee pointed out, both terms, GUVER(0) and GUVER, have been used for the UVER global irradiance at the Earth's surface. In order to homogenize the manuscript, only the term $G_{UVER,0}$ will be employed. Consequently, the Equation (1) has been rewritten. Additionally, the same criterion has been apply to the nomenclature of the total diffuse, total global and UVER diffuse irradiance.

**C6 (p. 5, Equation (3)):** r0 and r should be defined separately, not only via their ratio in line 148.

**R6:** These two magnitudes have been defined as follows (166-167):

*"where $\theta$ is the solar zenith angle, $r_0$ is the mean sun-earth distance, and r is the actual sun-earth distance for each date. The eccentricity correction-factor of the Earth's orbit, $(r_0/r)^2$, was calculated using the Spencer's formula [Spencer, 1971]."*

**C7 (p. 5, lines 152-155):** Figure 1 shows the relationship between UVER diffuse fraction (fUVER) and UVER transmissivity (kUVER). If I understand correctly the first one is derived from measurements of the K&Z radiometers (GUVER(0) and DUVER(0)) and the second one is derived from measurements of GUVER(0) and from a computation (Eq. 3). If it is true, the authors should mention all that (see also another comment below).

**R7:** Following the referee's comment, the text has been rewritten as follows (lines 174-180):

*"Figure 1 shows the relationship between UVER diffuse fraction ($f_{UVER}$) and UVER transmissivity ($k_{UVER}$), as derived from Equations (1) and (2), respectively. The former is derived from measurements of the K&Z radiometers ($G_{UVER,0}$ and $D_{UVER,0}$) and the latter is derived from measurements of $G_{UVER,0}$ and from a computation (Eq. 3). A general dependence can be clearly seen though the large scatter suggests the influence of other factors as well. In order to account for this variability, additional magnitudes directly related to the absorption and scattering of radiation in the atmosphere must be considered. This study focused on models relying on variables commonly available at standard radiometric stations, such as radiometric quantities and sun-geometry parameters directly related to the absorption and scattering of radiation in the atmosphere."*

**C8 (p. 5, line 156):** What does "meteorological magnitudes" mean ? Is it not rather "meteorological quantities" or "meteorological parameters" ?

**R8:** "Meteorological magnitudes" has been replaced by "meteorological quantities".

**C9 (p. 5, line 156-157):** I don't understand why the authors state that other parameters "could have been included in the models for estimating UVER diffuse fraction". As I told just above concerning Fig. 1, it sounds to me that $f_{UVER}$ was derived from measurements, not from modeling. I am rather confused, so the authors should clarify all that.

**R9:** Aimed to a clearer text, these lines have been deleted.

**C10 (p. 5, line 160):** Replace "radiometric magnitudes" with "radiometric quantities" or "radiometric parameters".

**R10:** "Radiometric magnitudes" has been replaced by "radiometric quantities".

**C11 (p. 7, Eq. (6) and (7))**: Define N.

**R11:** N is the number of cases within each hour. This informations has been added in line 243.

**C12 (p. 7, line 224):** The authors say "where sigma is the standard deviation", they should precise of what it is the standard deviation (of $k_{UVER}$).

**R12:** This points has been clarified.

**C13 (p. 10, Eq. (19) and (20)):** xi and xi* must be defined just after the equations, not much later in lines 325-326.

**R13:** The definition of xi* and xi has been written now just after Eqs. (19) and (20)

**C14 (p. 11, line 334):** Explain what "ordinary" means.

**R14:** Ordinary Least Squares (OLS) is another name for linear least squares. It has been clarified in the text as follows (line 359):

"Ordinary least squares (also known as linear least squares) fitting for models..."

**C15 (p. 11, lines 334-336):** The authors must explain why they have chosen different fittings (ordinary least squares and non-linear) for two groups of models.

**R15:** The use of the different fittings has been explained in the text as follows (330-334):

*"In principle, linear fitting is preferred since it requires no starting values of the fitting coefficients. Therefore, linear least squares fitting was applied whenever possible, that is, to models which are linear (REU, CGU1, CGU2, CGU3 and KUU) or linearizable, i.e. that can be reduced to a linear form with a change of variables (BOU and RIU). For the remaining cases (RAU1, RAU2 and RAU3) it was necessary to apply non linear fittings."*

**C16 (p. 12, lines 367-368 and Fig.3):** The relative residuals are averaged by intervals of each variable, these intervals should be specified. In Fig. 3 the caption should mention "Mean relative residuals of each UVER diffuse fraction. . .". Moreover dispersion bars around each mean are needed.

**R16:** Error bars showing the standard deviation of the mean have been added to Figure 3 and the caption of Figure 3 has been rewritten as the referee suggested to arrive at:

*"Figure 3. Mean relative residuals of each UVER diffuse fraction model versus a) solar zenith angle, b) UVER transmissivity, and c) predicted UVER diffuse fraction values. The range of the solar zenith angle (10°- 70°) has been divided into six equal bins while the ranges of the $k_{UVER}$ (0, 0.025) and $f_{UVER}$ (0,1) have been split into ten bins. The mean value for each bin (central point) and the standard deviation of the mean are shown (error bars)"*

**C17 (p. 12, line 381-382):** The authors state ". . . although the functional form can be generally suitable for other locations. . .", did they check that ? If yes they must give examples, if not they must reconsider their statement.

**R17:** Following the referee's comment this part of the sentence has been removed.

**C18 (p. 4, line 110):** Replace "without been affected" with "without being affected".

**R18:** This part of the text has been modified

**C19 (p. 6, line 184-185):** Replace "This new variable has been included by adding the a term. . ." with "This new variable has been included by adding a term" ("the" has been removed).

**R19:** The text has been corrected according to the suggestion given by the referee.

**C20 (p. 7, Eq. (8)):** In the left hand side term replace the superscript GC1 with GCU1.

**C21 (p. 7, Eq. (9)):** In the left hand side term replace the superscript GC2 with GCU2.

**C22 (p. 8, Eq. (10)):** In the left hand side term replace the superscript GC3 with GCU3.

**R20, R21 and R22:** These equations have been corrected

**C23 (p. 9, line 291):** "Ridley et al. [2008]" → "Ridley et al. [2010]".

**R23:** The reference has been corrected.

**C24 (p. 13, 422):** "models" → "model".

**R24:** The text has been corrected.

**C25:** The reference "Arola et al., 2003" cited p. 1, line 33 is missing.

**R25:** The corresponding reference has been included in the reference list as follows:

Arola A., K. Lakkala, A. Bais, J. Kaurola, C. Meleti, P. Taalas,. Factors affecting short- and long-term changes of spectral UV irradiance at two European stations, J. Geophys. Res., 108, 4549, 2003.  doi:10.1029/ 2003JD003447

**C26:** The reference "Craig et al., 2014" cited p. 2, line 40 is missing.

**R26:** The correct reference is: Williamson et al. [2014]. This mistake has been corrected.

This reference is included in the reference list as:

Williamson, C., R. Zepp, R. Lucas, S. Madronich, A. Austin, C. Ballare, M. Norval, B. Sulzberger, A. Bais, R. McKenzie, S. Robinson, D. Häder, N. Paul, and J. Bornman. Solar ultraviolet radiation in a changing climate, *Nature Climate Change*, 4, 434-441, 2014.

**C28 (Fig. 2):** It is difficult to distinguish the various colors red, magenta and brown. Moreover I cannot see the yellow dot. I suggest making different symbols of various colors.

**R28:** Colors and symbols size have been change in order to improve the quality of this Figure. The referee should be take into account that models REU (now grey color) and model KUU (now yellow color) present nearly the same behaviour and they overlap as in mentioned in line 391.

**C29 (Fig 3):** Replace the x-axis caption of the right plot "Fraciton" with "Fraction". Replace also "Relative residuals" with "Mean Relative residuals" on the y-axes.

**R29:** The text in Figure 3 has been corrected.

**C30 (Table 1):** The authors should add the number of cases for fitting and validation for each model.

**R30:** The number of cases for fitting (3979) and validation (1262) has been included in Table 1. This information has also included in Section 3.2.

---

## Author Response (AR2)

**Response to Co-Editor**

Comments to the Author:
5  The authors have addressed all reviewer's comments and the manuscript has been improved.
I have three minor comments that the authors could take into account for the final version of the manuscript.

**Comment a:**
The second reviewer is mentioning this and i do not think that there is a clear answer: Figure 1 needs some clarification on
10  what exactly are the data used.

**Response a:** We have tried to describe the data more clearly by being more explicit, as follows:

*"Figure 1 shows the relationship between UVER diffuse fraction ($f_{UVER}$) and UVER transmissivity ($k_{UVER}$), as derived from*
15  *Equations (1) and (2), respectively. Thus, the UVER diffuse fraction was obtained as the ratio between the measurements of*
*$D_{UVER,0}$ and $G_{UVER,0}$ performed by K&Z radiometers. On the other hand, the UVER transmissivity was calculated as the ratio*
*between the measurements of $G_{UVER,0}$ and the values of $G_{UVER,TOA}$ obtained by applying Equation (3)."*

20  **Comment b:**
On the discussion at page 1 line 34, i think that Zerefos et al., 2012 (https://www.atmos-chem-phys.net/12/2469/2012/)
provide an extended discussion on cloud, aerosol , ozone effects on solar radiation for a number of stations. In addition, den
Outer et al, 2005 (http://onlinelibrary.wiley.com/doi/10.1029/2004JD004824/pdf) compiled a great study of this aspect on
national level. I think both references could be added.
25

**Response b:** Both references have been  added (page 1, line 34; and list of references).

den Outer, P.N., H. Slaper, and R. B. Tax: UV radiation in the Netherlands: Assessing long-term variability and trends in
relation to ozone and clouds, Journal of Geophysical Research, 110, D02203, 2005. DOI:10.1029/2004JD004824, 2005.
30

Zerefos, C.S., K. TourpalI, K. Eleftheratos, S. Kazadzis, C. Meleti, U. Feister, T. Koskela, and A. Heikkilä: Evidence of a
possible turning point in solar UV-B over Canada, Europe and Japan, Atmos. Chem. Phys., 12, 2469–2477, 2012.

35  **Comment c:** page 4 , line 138 aerosol scattering versus wavelength is basic atmospheric physics so I do not think the
Kaskaoutis et al reference is appropriate. A more general refeence should be used.

**Response c:** The reference has been replaced by the more general reference Iqbal [1983], which was already in the list of
references.
40
* * *
We would like to thanks the referres for their  valuable comments that have improved this manuscript. For that reason, we
have been included the following sentence in the Acknowledgements section:
45

*"Thanks to the referees for their comments and suggestions, which notably improved this paper."*